# Agreement between Accelerometer-Assessed and Self-Reported Physical Activity and Sedentary Behavior in Female Breast Cancer Survivors

**DOI:** 10.3390/diagnostics13223447

**Published:** 2023-11-15

**Authors:** Malgorzata Biskup, Pawel Macek, Malgorzata Terek-Derszniak, Marek Zak, Halina Krol, Krzysztof Falana, Stanislaw Gozdz

**Affiliations:** 1Collegium Medicum, Jan Kochanowski University, Avenue IX Centuries Kielc 19A, 25-516 Kielce, Poland; pawel.macek@onkol.kielce.pl (P.M.); mzak1@onet.eu (M.Z.); krolhalina@poczta.fm (H.K.); stanislaw.gozdz@onkol.kielce.pl (S.G.); 2Department of Rehabilitation, Holycross Cancer Center, Artwinskiego 3 Street, 25-734 Kielce, Poland; m.terek@poczta.fm; 3Department of Epidemiology and Cancer Control, Holycross Cancer Center, Artwinskiego 3 Street, 25-734 Kielce, Poland; 4Department of Research and Education, Holycross Cancer Center, Artwinskiego 3 Street, 25-734 Kielce, Poland; 5Faculty of Law and Social Sciences, Jan Kochanowski University, Uniwersytecka Street 15, 25-406 Kielce, Poland; krzysztof.falana@onkol.kielce.pl; 6Department of Clinical Oncology, Holycross Cancer Center, Artwinskiego 3 Street, 25-734 Kielce, Poland

**Keywords:** breast cancer, physical activity, accelerometer, IPAQ, ActiGraph GT3X-BT, radiotherapy, chemotherapy

## Abstract

An accurate quantitative assessment of physical activity and sedentary lifestyles enables a better understanding of their relationship with the health records of cancer survivors. The objective of this study was to compare the subjective and objective methods of physical activity measurement in female breast cancer survivors. Materials and methods: In total, 135 female breast cancer survivors at the Holycross Cancer Center, Kielce, Poland, were included in this study. A shortened version of the International Physical Activity Questionnaire (IPAQ) was used to subjectively assess the participants’ physical activity (PA), and an ActiGraph GT3X-BT accelerometer was used for an objective assessment. In total, 75% of the studied women did not report any vigorous PA, irrespective of the measurement method. The average values of moderate PA and moderate-to-vigorous PA (MVPA) measured with IPAQ compared with the accelerometer were sevenfold and tenfold higher, respectively. Conversely, the sedentary behavior values measured with the accelerometer were almost three times higher than those measured with IPAQ. The PA and sedentary behavior measurements were significantly different. Irrespective of PA intensity, the accelerometer-based measurements produced significantly lower results than IPAQ, while higher results were observed for sedentary behavior. The measurement differences between these two methods increased as the average differences grew. Regardless of the measurement method, a negative association was observed between moderate PA with general adiposity and adipose tissue distribution, whereas sedentary behavior demonstrated an opposite trend. This indicates the detrimental role of obesity in limiting PA.

## 1. Introduction

There is much scientific evidence confirming the positive influence of physical activity (PA) on reducing the risk of cancer development or recurrence. PA has also been proven to have a positive influence on recovery among breast cancer survivors. Researchers suggest that PA is safe both during and after cancer treatment [1,2,3,4,5,6,7,8].

According to the American Cancer Society, 150 min of moderate exercise or 75 min of vigorous exercise a week is recommended to reduce the chance of developing cancer [9]. For cancer patients, because of the common occurrence of multimorbidity and advanced age, moderate exercise lasting at least 30 min a day, five days a week, is recommended [10]. Reducing sitting time is also worth investigating. It is worth pointing out that a sedentary lifestyle differs from a complete lack of activity, as a person who follows active lifestyle recommendations can still lead a sedentary lifestyle [11,12]. Sedentary behaviors include all activities during waking hours in which a sitting or lying position predominates and energy expenditure does not increase [6,13,14]. The benefits of PA and reduced sedentary behavior are undeniable. The main challenge for researchers, however, remains the selection of proper measurement methods [15].

Most research employs self-assessment questionnaires, which are based on the examined person recalling their PA and/or sedentary lifestyle [15,16,17,18]. However, reported PA (mainly moderately and vigorously intensive) is usually overestimated, and sitting hours are underestimated [12]. Moreover, recording light exercise and sitting time can be difficult [12,15].

One of the methods by which researchers can cope with the limitations of self-assessment is using an objective measure of physical exercise and sedentary behaviors, such as accelerometry. It has been proven that agreement between the mentioned techniques (questionnaires and accelerometry) is weak or, at most, satisfactory [12,15,19].

Accelerometers register the participant’s overall PA and sedentary time over 24 h. They collect data about activity duration, intensity, and frequency and, at the same time, limit errors and unavoidable deviations often seen in subjective measurements [12]. Detailed quantitative assessments of PA and sedentary lifestyles utilizing accelerometry provide a better understanding of these behaviors and their associations with the health records of cancer survivors. Accelerometer data also enable a thorough analysis of PA accumulation patterns, including the times of day and days of the week when people are more or less active or lead a sedentary lifestyle. This information is valuable for developing interventions aimed at changing these behaviors [15,20]. 

Although accelerometers allow for delivering more abundant and precise data on PA and sedentary behaviors than self-assessments, the quality of the produced data depends on a number of decisions made during data collection and processing. These critical decisions include the type of device; its positioning; device-wearing time protocols; epoch duration; filter application; idle time criteria; valid device wear time; and the way in which the data are processed to determine the total sitting time, the length of light activity, and the length of moderate-to-vigorous physical activity (MVPA) [15,21,22]. 

Some of the first population-level studies containing objectively measured PA and sedentary behavior were the studies conducted by the National Health and Nutrition Examination Survey (NHANES) in the years 2003–2004 and 2005–2006, which allowed for a better understanding of these behaviors [12]. 

The main objective of this research paper was to compare the subjective (IPAQ) and objective (accelerometer) levels of PA and sedentary behavior in female breast cancer survivors. We assessed whether body mass has an influence on PA levels and sedentary behavior. 

The secondary objectives were as follows:-To compare the average levels of moderate PA and moderate-to-vigorous physical activity (MVPA) as measured using IPAQ and accelerometry;-To compare the average sedentary behavior values measured utilizing IPAQ and accelerometry;-To investigate the associations between PA, sedentary behavior, general adiposity, and adipose tissue distribution.

## 2. Materials and Methods

### 2.1. Study Population

This original epidemiological clinical study comprises a cohort of 135 female breast cancer survivors at the Holycross Cancer Center in Kielce, Poland. The study was approved by the Ethics Committee in Kielce on 19 May 2017 (approval no. 19/2017). The participants were fully informed about the examination method and provided signed consent. The entire study was carried out in the Department of Rehabilitation of Holycross Cancer Center (in year 2022). All patients visiting the rehabilitation outpatient clinic were invited to participate in the study. Only patients who provided informed consent for participation were included in the study.

The inclusion criteria for this study were as follows: a confirmed diagnosis of breast cancer, the completion of surgical intervention (unilateral or bilateral mastectomy), female sex, a stable clinical state, and the completion of radiation therapy and/or chemotherapy. The exclusion criteria included male sex and hospitalization on the day of assessment. 

### 2.2. Demographics and Cancer Treatment Variables

A questionnaire involving demographic information (age, educational background, marital status, area of residence) and medical data (cancer risk factors, treatment history, comorbidities) was used in the study. Anthropometric measurements included body mass, height (self-reported by the participants), and waist (WC) and hip circumferences (measured by the personnel).

### 2.3. General Adiposity Assessment

The body mass index (BMI) of the study participants was computed using anthropometric data, specifically body mass (kg) divided by the square of height (m²). The values of BMI were stratified into three categories based on the World Health Organization (WHO) guidelines: normal weight was defined as a BMI of 18.50–24.99 kg/m^2^, underweight as a BMI of less than 18.50 kg/m^2^, overweight as a BMI of 25.00–29.99 kg/m^2^, and obesity as a BMI ≥ 30.00 kg/m^2^ [23].

### 2.4. Adipose Tissue Distribution Measurements

The measurements of waist and hip circumferences were taken using a metric, non-elastic measuring tape, positioned parallel to the floor and recorded to an accuracy of 0.1 cm, in line with anthropometric measurement guidelines [24,25,26]. In accordance with the WHO protocols, WC was measured at the midpoint between the iliac crest and the lower border of the last palpable rib, in the mid-axillary line. Hip circumference was measured at the level of the largest lateral extension of the hips. 

A normal WC was categorized as lower than 88 cm, and central obesity was defined as ≥88. WHR was calculated by dividing the WC by the hip circumference. The values were then classified as “normal” (WHR < 0.80) or “abdominal obesity” (WHR ≥ 0.80). WHtR was calculated by dividing an individual’s WC by their height.

### 2.5. Sedentary Behavior and Physical Activity Assessment

A subjective assessment of PA was performed using a shortened version of the International Physical Activity Questionnaire (IPAQ). Objective PA was gauged utilizing an ActiGraph GT3X-BT accelerometer with ActiLife 6 software. 

The short IPAQ form included 7 questions addressing PA associated with daily living, occupational duties, and leisure time activities. Actions conducted in the workplace, within the home environment, in transit, and during recreation, as well as exercise or sports, were taken into consideration. Sitting, walking, and PA (vigorous or moderate) times were collected. Only activities lasting longer than 10 minutes, without interruption, were recorded [27].

As per IPAQ methodology, the participants were divided according to their total PA levels into three categories: low, moderate, and high. 

-Low-level activity—individuals who do not meet the criteria for the other two categories, with PA at a level of <600 metabolic equivalents of task (MET)-min/week.-Moderate-level activity—PA at a level of 600–1500 MET-min/week, or 1500–3000 MET-min/week, although with 1 or 2 days comprising high-intensity exercise.-High-level activity—1500 MET-min/week, although with at least 3 days comprising high-intensity exercise, at over 3000 MET-min/week [28].

A well-validated, triaxial ActiGraph GT3X-BT accelerometer (Pensacola, FL, USA), able to provide objective PA measurements, was used to measure the frequency, duration, and intensity of sedentary behavior, light PA, and MVPA. The accelerometer has a built-in inclinometer, which records the participant’s position in three dimensions; therefore, it was possible to differentiate sitting and standing positions. The participants were instructed to wear the accelerometer at waist level for 7 days, 24 hours a day. Upon completion of this duration, the participants returned the devices and were provided with feedback along with an activity measurement printout.

All complete and valid datasets were processed in ActiLife 6 software using a low-frequency extension and aggregated to 60 s epochs. Every minute of device wear time was classified by intensity (counts per minute, cpm), utilizing Freedson cut points as follows: <100 cpm, sedentary lifestyle; 100–1.951 cpm, LPM; and 1.952–5.724 cpm, MVPA ≥ 5.725 cpm [12]. 

Wear time was assessed using ActiLife 6 software, based on the Troiano 2007 algorithm. The minimum threshold for wear time was established as 3 days. Non-wear time was defined as periods of 60 or more minutes of continuous 0 counts, with an allowance of up to 2 min of observation of less than 100 cpm [29].

For each valid day on which the accelerometer was worn, the numbers of minutes classified as sedentary lifestyle, light PA, or MVPA were recorded. These values provided estimates of the total duration devoted to each activity throughout the day. The daily estimates for each activity were then averaged across all valid days per participant at every assessed time point, thereby yielding the daily average minutes spent on each activity. The number of minutes in each category was divided by the total device wear time, producing an estimate percentage of the day spent on the given behavior [30,31].

### 2.6. Statistical Analyses

Descriptive statistics are presented as the mean with standard deviation, median with interquartile range (IQR), range (minimum to maximum), or as a number and proportion, depending on the characteristics of the studied variable. Statistical differences between the accelerometer and IPAQ results were studied with the Wilcoxon signed-rank test. The concordance between the two measurement methods is illustrated via Bland–Altman plots. The relationships between PA and anthropometric measurements (BMI, WC, WHR, and WHtR) were examined utilizing robust regression models. Unadjusted and adjusted models were fitted. The model was adjusted for age, area of residence, marital status, education level, occupational status, and comorbidities. *p*-values of <0.05 were considered statistically significant. All statistical analyses were performed using R software, version 3.6.3.

## 3. Results

This study included 135 female breast cancer survivors. The average age was 63.2 ± 10.0 years. All women underwent surgical treatment. A total of 29.6% (*n* = 40) of the participants received one method of cancer treatment, 35.6% (*n* = 48) underwent two, and 34.8% (*n* = 47) were treated with three methods (Table 1). More than half of the women (63%) resided in urban areas, two-thirds were in a relationship, and 85% had university degrees. Moreover, 70% of the studied women reported no professional activity at the time of the survey, and approximately 70% had comorbidities.

Among the participants, 34.8% (*n* = 47) had a normal BMI, and 27.4% (*n* = 37) were classified as obese (Table 2). Notably, 75% of the women studied did not report any engagement in vigorous PA, a finding consistent across both measurement methods. When assessed using IPAQ, the average values of moderate PA and MVPA were 7 and 10 times higher than those reported by the accelerometer, respectively. In contrast, self-reported sedentary behavior was threefold lower than the sedentary behavior reported by the accelerometer.

A significant discrepancy was observed between the measurements of PA and sedentary behavior when utilizing the accelerometer and IPAQ (Table 3). IPAQ overestimated the average values of moderate PA by 160.7 min per day (*p* < 0.001), vigorous PA by 8.0 min per day (*p* < 0.05), and MVPA by 234.6 min per day (*p* < 0.001). Conversely, the average sedentary behavior reported by IPAQ was lower by 500.9 min per day (*p* < 0.001) compared to the accelerometer data. The negative bias for moderate PA, vigorous PA, and MVPA was accentuated, exceeding 50 min per day (Figure 1).

For all activity types examined, the confidence intervals for the average differences did not include zero, pointing to the significance of the observed biases (Figure 1). Considerable widths of the limits of agreement corresponded to ambiguous measurement results. Notably, for moderate PA, vigorous PA, and MVPA, a trend was observed, indicating that measurement differences increased as the average difference increased.

The unadjusted regression models revealed that MVPA, as measured with both the accelerometer and IPAQ, was significantly inversely related to WC, WHtR, and WHR. The negative regression coefficients suggest that a decrease in MVPA corresponded to an increase in anthropometric measurements (Table 4). The relationships between sedentary behavior and the anthropometric features studied (with the exception of WHR, as measured by the accelerometer) were not significant, regardless of the measurement method used (*p* > 0.05). However, positive values of the regression coefficients indicate that an escalation in anthropometric measurements may correspond to an increase in sedentary behavior.

After the adjustment of the regression models to account for age, area of residence, marital status, education level, occupational status, and comorbidities, a significant relationship between IPAQ-assessed sedentary behavior and BMI (*p* = 0.0166), WC (*p* = 0.0236), and WHtR (*p* = 0.0214) was observed (Table 5). The aforementioned relationship was positive, confirming the role of obesity in limiting PA. 

## 4. Discussion

Our study included 135 female breast cancer survivors. We evaluated their PA utilizing both an objective (accelerometer) and subjective (IPAQ) method. The results of our study confirm that female breast cancer survivors tend to avoid vigorous PA—it was not recorded in half of the participants, irrespective of the measurement method employed. Based on both the existing literature and our clinical experience with cancer patients, we postulate that this reluctance could stem from concerns about the possibility of upper extremity overload after lymphadenectomy, as well as experiencing fatigue that is often attributable to cancer treatment [32,33,34]. Although it has been established that cancer patients can safely perform high-intensity activities, which confer health benefits beyond conventional exercise regimens, this group of patients seemed to hesitate about undertaking vigorous PA [35]. On the contrary, the fatigue and pain related to cancer likely contribute to overall disease- and treatment-related burden. This makes it challenging for patients, especially those in more advanced disease stages, to participate in intensive PA [36]. 

It is worth noting that our research did not include a comparative analysis with healthy individuals. Kang et al., who conducted such an analysis, observed higher levels of PA among healthy individuals than female breast cancer survivors. They demonstrated statistically significant differences in the levels and frequency of PA and METs. The results of vigorous PA, moderate PA, and light PA in patients with breast cancers were significantly lower than in the healthy population [37].

Our study further examined PA measured with an objective method, via accelerometry. The application of this technique in assessing PA appears to be fundamental to understanding the problem of PA among cancer patients. This method provides insights that enable us to optimize physiotherapeutic interventions and promote PA effectively. Notably, the National Health and Nutrition Examination Survey (NHANES), conducted between the years 2003 and 2004 and 2005 and 2006 and published in 2017, was the first study analyzing both PA and sedentary behavior using objective measures on a representative sample of people who survived different types of cancer [12]. According to the study, cancer survivors spent 36% of their waking time on light PA, 2% on MVPA, and 62% on sedentary behavior. Moreover, the studied group spent 307 minutes per day on light PA and 16 minutes per day on MVPA, and only 8% adhered to the WHO PA guidelines. The studied population reported, on average, 519 minutes of sedentary behavior, with 86 breaks in sedentary behavior throughout the day. These results indicate that people who have survived cancer are not sufficiently active [12].

Our results demonstrate that the IPAQ-reported moderate PA and MVPA values were, respectively, sixfold and sevenfold higher than the accelerometry values. In contrast, the sedentary behavior values measured with the accelerometer were almost three times higher than those reported through self-assessment. Colley et al. found that the levels of PA self-reported by adult Canadians were higher than those measured by accelerometers. Overall, a comparison of self-reported and objectively measured activity levels revealed a low correlation and agreement [38].

In turn, Ortiz et al. reported a moderate, yet statistically significant, positive relationship between MVPA measured via IPAQ (138 MET minutes per day) and via the Actigraph accelerometer (23 minutes per day). Irrespective of the positive relationship between these subjective and objective measurement methods, IPAQ-assessed MVPA was sixfold overestimated compared to the accelerometer readings [39].

Lee, Macfarlane, Lam, and Stewart conducted a systematic review of 23 research papers, validating IPAQ in comparison to activity monitors [40]. They reported a range of 28% to 173% overestimation of self-assessed PA. These results not only demonstrate the limitations of self-assessed PA measurements but also indicate the notably low level of activity among Latina breast cancer survivors [39].

While addressing the problem of low PA, the concurrent challenge of a sedentary lifestyle should not be overlooked. Our results indicate that the women treated for breast cancer assessed their sitting time to be 279.9 minutes per day. When measured objectively, their average daily sitting time was 782.7 minutes. 

Numerous studies describe interventions aimed at increasing MVPA rates in cancer survivors [41,42,43,44,45]. However, few studies have examined the impact of increased PA on sedentary behavior.

In a 12-week, randomized controlled study investigating a peer-led intervention aimed at increasing the PA of 76 breast cancer survivors, the intervention successfully amplified their PA levels. However, the intervention did not significantly reduce sitting time, either within the group or when compared to the control group [46]. These findings imply that PA-related interventions may not necessarily decrease sedentary time in this demographic. The lack of effect of the intervention in the study conducted by Pinto et al. could be attributable to a compensatory “rest and recover” mechanism, where higher activity leads to fatigue; therefore, sedentary time remains the same or even increases in periods of inactivity [46,47]. The post-MVPA “rest and recover” phenomenon has been documented in the intervention literature in healthy populations, but it is insufficiently studied among cancer survivors. This process could be particularly important for breast cancer survivors, who often report experiencing fatigue post-treatment. Given the growing evidence linking a sedentary lifestyle to adverse health consequences in cancer survivors, it is crucial to understand how activity-related interventions influence both PA and sedentary behavior [11].

The research conducted by Maridaki et al. involved a cohort of women undergoing chemotherapy for breast cancer. It demonstrated that, although these patients engaged in physical exercise, they also accumulated substantial daily sitting hours, both at work and at home. This phenomenon was attributed to the cessation of work during chemotherapy, leading patients to spend more time sitting at home, potentially contributing to weight gain. The authors corroborated that a higher BMI is related to a sedentary lifestyle after the diagnosis of cancer. Since an increased body mass has been associated with an increased risk of disease recurrence and lower survival rates, all cancer patients should not only avoid physical inactivity but should also adhere to specific exercise recommendations to optimize the health benefits from the activity [36]. Both increasing PA and decreasing sedentary behavior are vital to maintaining the physical fitness of female breast cancer survivors.

Our research demonstrates a relationship between anthropometric measurements and PA levels, which are inseparable from physical fitness and are vital for female breast cancer survivors [48]. With an increase in body mass, PA levels tend to decrease. 

Numerous studies confirm that cancer treatment results in a decrease in PA, fatigue, and weakness. The type of cancer treatment, nutritional status, and PA level all influence changes in body mass [49,50,51,52,53,54,55]. It has been established that undertaking PA after a breast cancer diagnosis decreases fatigue and other adverse effects of cancer treatment, along with improving the life quality and overall fitness of cancer survivors [56].

Overweight and obesity in female breast cancer survivors may stem from an imbalance between calorie intake and energy expenditure due to metabolic alterations [57]. These metabolic changes are often a consequence of menopause and the associated hormonal shifts induced by treatment. During menopause, estrogen levels decrease, leading to an accumulation of adipose tissue in the hip and waist areas [58,59]. Moreover, evidence suggests that BMI has only a marginal influence on the timing of cancer diagnosis [60,61,62]. 

The American Cancer Society has published guidelines on nutrition and PA for cancer survivors [63]. Moreover, the National Comprehensive Cancer Network has incorporated nutrition, body weight control, and PA into their survival-related guidelines [64]. Yet, including these factors in survivor care poses a challenge, with the limited availability of research and information on best practice in this area. Healthcare providers encounter numerous barriers when addressing these issues, including limited knowledge and training, as well as uncertainty about the optimal timing for introducing survivorship care plans within the disease continuum. They also complain of a lack of administrative support and problems resulting from limited time and patient overload [65,66]. Support for cancer survivors can take on various forms, and treatment outcomes are influenced by multiple factors [56,67]. It is crucial to provide patients with proper education. There is also the possibility of using the American Society of Clinical Oncology booklet Managing Your Weight After a Cancer Diagnosis: A Guide for Patients and Families (www.cancer.net/sites/cancer.net/files/weight_after_cancer_diagnosis.pdf, accessed on 7 June 2023), the American Cancer Society website (www.cancer.org/treatment/survivorship-during-and-after-treatment/staying-active.html, accessed on 7 June 2023), or the National Comprehensive Cancer Network websites (www.nccn.org/patients/resources/life_after_cancer/nutrition.aspx and www.nccn.org/patients/resources/life_after_cancer/exercise.aspx, accessed on 7 June 2023) [56].

The mentioned data are critical when considering the increased risks of mortality and disease recurrence [53]. The relationship between body mass at the time of breast cancer diagnosis and the risk of recurrence or death has been assessed in more than 100 research papers over the last 40 years [56,68,69,70]. A meta-analysis from 2014, encompassing 82 studies involving a total of 213,075 women with early-stage breast cancer, reported a 35% increase in cancer-related mortality and a 41% increase in overall mortality rates in women who were obese at the time of diagnosis, compared to individuals with a normal body mass [56].

In clinical practice, it is vital to inform patients who have survived breast cancer about the significance of PA in its various forms and intensities. Moreover, encouraging them to increase their PA, irrespective of the time that has elapsed since their surgery, could be an important factor in preventing recurrence, reducing obesity, and enhancing overall well-being [62].

A crucial role in the education of patients and their families is attributed to the physician and the physiotherapist. At the Holycross Cancer Center, where this research was conducted, patients receive instructions on anti-swelling prophylaxis and a physical exercise tutorial one day prior to surgery. This educational initiative, along with exercise and self-massage tutorials, is continued throughout the patient’s hospital stay. Additionally, patients receive an informative booklet containing essential guidance. Implementing a program to monitor physical activity in specific periods of time could prove beneficial. In the case of reduced PA, the physician and physiotherapist could interview the patient in search of the cause of this situation and intervene if necessary. It is our conclusion that PA levels depend not only on the education provided by the healthcare personnel but also on the patient’s exercise habits before their cancer diagnosis. Patients’ well-being during various phases of treatment also holds significant importance. Our future research will focus on the PA of breast cancer survivors, examining how it varies over time following surgery and in response to different treatment modalities.

### Strengths and Weaknesses of the Study

Our study has several limitations. The accelerometer does not capture certain types of PA, such as biking or swimming. It may register signals that, in fact, are not activities (for example, it can misidentify sitting in a car as movement and not as sedentary behavior). Furthermore, although the Freedson cut points are widely used to assess PA in cancer survivors, they are derived from healthy, young adults with an average age of 24 years [29]. As a result, the accelerometer may not register optimal MVPA in older populations, including cancer survivors [71]. The studied groups often present with comorbidities, as well as long-lasting symptoms of cancer, which influence their functional performance. Therefore, the cut-off points derived from a young and healthy population will not always be appropriate for cancer survivors [71]. Accelerometers are reliable and objective PA measuring devices but are also more expensive, and data interpretation is more time consuming than self-report methods. 

Nevertheless, this study also has multiple strengths. We utilized validated, robust measurement methods. The use of accelerometry to measure sitting significantly reduces the recall and response biases associated with self-assessment methods. Furthermore, accelerometers are also better at recording light PA and sedentary behavior. The ability to precisely assess light PA is crucial in this population, because, as the data indicate, this population tends to engage in more light PA than MVPA.

Another strength of our study is the combined use of two measurement methods—an accelerometer and IPAQ [72]. In the case of IPAQ, the methodological issues are particularly related to the surveying technique, classification, and duration of activity, as well as the timing of assessments. The questionnaire has been developed to allow for international comparisons, and it has been extensively validated. However, the imprecise, self-reported values limit its usability in research compared to objective measures, such as accelerometry [73,74].

A further strength of our study is the interviewer-administered surveying technique. According to international regulations, questionnaires can be completed in-person by the respondent or via telephone interviews. The self-administration of questionnaires leads to considerable overestimations of the type and duration of PA. It can also lead to the wrong classification of PA (usually classifying moderate PA as vigorous PA) [27].

In conclusion, the combined use of objective and subjective measurements of PA to describe these behaviors more precisely is crucial to understanding cancer survivors. The findings suggest that cancer survivors lead a sedentary lifestyle. The mentioned data align with those of previous research on breast and prostate cancer survivors [12]. 

The agreement between IPAQ and the accelerometer in evaluating PA levels was found to be low. This indicates that these two methods measure distinct PA constructs and cannot be used interchangeably to assess daily PA in cancer patients. Although we have various methods at our disposal, there remains no gold standard for measuring daily PA on a large scale.

## 5. Conclusions 

-Across all types of PA, measurement via accelerometry produced significantly lower results than IPAQ, with the average moderate PA and MVPA values being seven times and ten times lower, respectively.-The accelerometer-measured sedentary behavior values were nearly triple those measured using IPAQ. These measurement differences were consistent with the magnitude of the average differences observed. The association between moderate PA and both overall adiposity and adipose tissue distribution was negative, regardless of the measurement method used. Conversely, sedentary behavior exhibited a positive correlation with these variables, suggesting an adverse impact of obesity on physical activity levels.-The IPAQ method was subject to error in the assessment of PA, further increasing with higher objective activity values. However, the relationships with the detrimental effects of a lack of activity maintained the appropriate direction, similar to the accelerometer-based observations.-The observed limitation of vigorous PA is a consequence of breast cancer treatment.

## Figures and Tables

**Figure 1 diagnostics-13-03447-f001:**
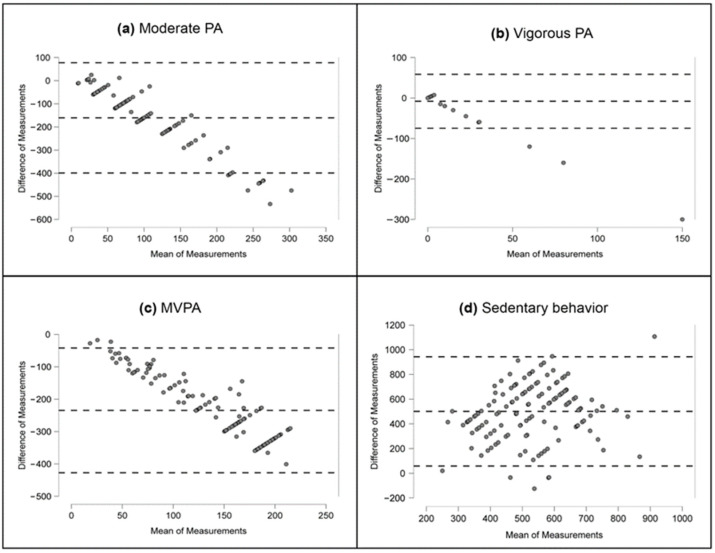
Differences between moderate PA, vigorous PA, MVPA, and sedentary behavior based on accelerometer and IPAQ. (**a**) Moderate PA; (**b**) Vigorous PA; (**c**) MVPA; (**d**) Sedentary behavior. Abbreviations: PA, physical activity; MVPA, moderate-to-vigorous physical activity. Note: horizontal lines are drawn at mean difference and at the limits of agreement, which are defined as the difference ±1.96 times the standard deviation of the differences.

**Table 1 diagnostics-13-03447-t001:** Basic characteristics of study group based on factor variables.

Characteristic	*n* (%)
Mastectomy side	
Left side	70 (51.9)
Right side	48 (35.6)
Both side	17 (12.6)
Lymphadenectomy	
No	82 (60.7)
Yes	53 (39.3)
RTH	
No	66 (48.9)
Yes	69 (51.1)
CHTH	
No	62 (45.9)
Yes	73 (54.1)
Area of residence	
Rural	50 (37.0)
Urban	85 (63.0)
Marital status	
In a relationship	90 (66.7)
Single	45 (33.3)
Education	
Higher level	115 (85.2)
Lower level	20 (14.8)
Occupational status	
Professionally active	40 (29.6)
Professionally inactive	95 (70.4)
Comorbidities	
No	38 (28.2)
Yes	97 (71.9)

Abbreviations: RTH, radiotherapy; CHTH, chemotherapy.

**Table 2 diagnostics-13-03447-t002:** Basic characteristics of study group based on continuous variables.

Characteristic	Mean (SD)	Median (IQR)	Min–Max
Age (years)	63.2 (10.0)	65.0 (14.0)	40.0–85.0
BMI (kg/m^2^)	27.5 (5.0)	26.8 (7.2)	18.0–46.7
WC (cm)	94.9 (13.2)	94.0 (19.0)	62.0–139.0
WHR	0.9 (0.1)	0.9 (0.1)	0.7–1.0
WHtR	0.6 (0.1)	0.6 (0.1)	0.4–0.9
Accelerometer (min/day)			
Moderate PA	23.2 (18.6)	19.6 (21.9)	0.1–95.3
Vigorous PA	0.2 (0.9)	0.0 (0.0)	0.0–7.5
MVPA	23.6 (18.8)	19.6 (22.4)	0.1–95.3
Sedentary behaviors	780.8 (185.8)	820.8 (317.8)	260.0–1466.8
Light PA	299.6 (76.2)	303.5 (98.8)	93.4–477.4
Steps (number)	6177.7 (2269.1)	6083.6 (3145.6)	919.6–13911.3
Energy expenditure	2344.7 (1226.0)	2153.7 (1458.0)	247.9–6153.3
IPAQ (min/day)			
Moderate PA	183.9 (124.7)	150.0 (120.0)	15.0–540.0
Moderate PA (MET)	4925.3 (3650.3)	3360.0 (4800.0)	60.0–15,120.0
Vigorous PA	8.2 (34.0)	0.0 (0.0)	0.0–300.0
Vigorous PA (MET)	136.9 (497.8)	0.0 (0.0)	0.0–2880.0
MVPA	258.2 (100.2)	300.0 (180.0)	32.1–411.4
Sedentary behaviors	279.9 (151.1)	240.0 (180.0)	30.0–800.0
Sum of MET	6670.8 (2594.1)	7518.0 (4746.0)	753.0–12,078.0
Follow-up	7.7 (7.3)	4.0 (10.0)	1.0–41.0

Abbreviations: BMI, body mass index; WC, waist circumference; WHR, waist-to-hip ratio; WHtR, waist-to-height ratio; PA, physical activity; MVPA, moderate-to-vigorous physical activity; IPAQ, International Physical Activity Questionnaire; MET, metabolic equivalent of task.

**Table 3 diagnostics-13-03447-t003:** Comparison between the accelerometer and IPAQ measurements for moderate physical activity, vigorous physical activity, MVPA, and sedentary behavior.

Characteristic	Bias and Limits	Point Value	Lower 95% CI	Upper 95% CI	*p*
Moderate PA	Mean difference + 1.96 SD	77.9	42.0	113.7	<0.001
	Mean difference	−160.7	−181.4	−140.0	
	Mean difference − 1.96 SD	−399.3	−435.1	−363.4	
Vigorous PA	Mean difference + 1.96 SD	58.6	48.6	68.6	<0.05
	Mean difference	−8.0	−13.8	−2.3	
	Mean difference − 1.96 SD	−74.7	−84.7	−64.6	
MVPA	Mean difference + 1.96 SD	−41.9	−70.9	−12.9	<0.001
	Mean difference	−234.6	−251.4	−217.9	
	Mean difference − 1.96 SD	−427.4	−456.4	−398.4	
Sedentary behavior	Mean difference + 1.96 SD	943.0	876.5	1009.5	<0.001
	Mean difference	500.9	462.5	539.3	
	Mean difference − 1.96 SD	58.8	−7.7	125.3	

Abbreviations: PA, physical activity; MVPA, moderate-to-vigorous physical activity; SD, standard deviation; 95% CI, 95% confidence interval.

**Table 4 diagnostics-13-03447-t004:** Unadjusted robust regression models for relationships of measures of overall adiposity and adipose tissue distribution with moderate-to-vigorous physical activity and sedentary behaviors.

Characteristic	Accelerometer		IPAQ	
	Estimate (95% CI)	*p*	Estimate (95% CI)	*p*
MVPA				
BMI (kg/m^2^)	−0.35 (−0.90, 0.19)	0.2195	−1.00 (−4.47, 2.47)	0.5728
WC (cm)	−0.24 (−0.45, −0.04)	0.0207	−1.17 (−2.49, 0.16)	0.0851
WHR	−47.53 (−92.28, −2.77)	0.0377	−337.12 (−619.34, −54.89)	0.0206
WHtR	−42.63 (−73.49, −11.79)	0.0091	−169.68 (−370.66, 31.31)	0.0993
Sedentary behaviors				
BMI (kg/m^2^)	2.72 (−3.36, 8.81)	0.3794	4.74 (−0.54, 10.01)	0.0789
WC (cm)	0.54 (−1.80, 2.88)	0.6514	1.86 (−0.10, 3.81)	0.0619
WHR	−85.90 (−591.02, 419.21)	0.0205	220.90 (−223.97, 665.77)	0.3300
WHtR	79.67 (−275.78, 435.12)	0.6619	245.98 (−57.34, 549.30)	0.1105

Abbreviations: BMI, body mass index; WC, waist circumference; WHR, waist-to-hip ratio; WHtR, waist-to-height ratio; IPAQ, International Physical Activity Questionnaire; 95% CI, 95% confidence interval.

**Table 5 diagnostics-13-03447-t005:** Adjusted robust regression models for relationships of measures of general adiposity index and fat distribution with MVPA and sedentary behaviors.

Characteristic	Accelerometer		IPAQ	
	Estimate (95% CI)	*p*	Estimate (95% CI)	*p*
MVPA				
BMI (kg/m^2^)	−0.01 (−0.60, 0.57)	0.9616	−0.58 (−4.50, 3.34)	0.7712
WC (cm)	−0.09 (−0.31, 0.13)	0.4434	−0.96 (−2.43, 0.51)	0.1984
WHR	−42.46 (−87.31, 2.38)	0.0549	−300.87 (−607.11, 5.37)	0.0549
WHtR	−10.78 (−45.34, 23.76)	0.5496	−115.05 (−350.78, 120.67)	0.3382
Sedentary behaviors				
BMI (kg/m^2^)	3.50 (−3.25, 10.26)	0.3113	6.55 (1.26, 11.84)	0.0166
WC (cm)	0.93 (−1.68, 3.54)	0.4876	2.46 (0.34, 4.59)	0.0236
WHR	6.89 (−542.37, 528.60)	0.0549	293.40 (−139.26, 726.07)	0.1859
WHtR	164.52 (−241.60, 570.63)	0.4316	379.35 (58.48, 700.21)	0.0214

Abbreviations: BMI, body mass index; WC, waist circumference; WHR, waist-to-hip ratio; WHtR, waist-to-height ratio; IPAQ, International Physical Activity Questionnaire; 95% CI, 95% confidence interval.

## Data Availability

The data presented in this study are available on request from the corresponding author.

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
