# Peer review of "Agreement between Accelerometer-Assessed and Self-Reported Physical Activity and Sedentary Behavior in Female Breast Cancer Survivors"

_diagnostics, 2023, doi:10.3390/diagnostics13223447_

Round 1
Reviewer 1 Report
Comments and Suggestions for Authors
I’ve read with attention the paper of Biskup et al. that is potentially of interest. The background and aim of the study have been clearly defined. The methodology applied is overall correct, the results are reliable and adequately discussed. I’ve only some minor comments:
- The abstract should contain some main quantitative data
- A graphical abstract could improve the attractiveness of the paper
- The authors should stress among limitation that the study has been carried out in a specific country and setting, so that the results obtained can not immediately inferred to different settings and countries.
Author Response
I ’ve read with attention the paper of Biskup et al. that is potentially of interest. The background and aim of the study have been clearly defined. The methodology applied is overall correct, the results are reliable and adequately discussed. I’ve only some minor comments:
- The abstract should contain some main quantitative data
We are grateful for the thorough review and directions concerning our paper.
We added additional quantitative data in the abstract: “75% of studied women did not report any vigorous PA, irrespective of measurement method.” “Conversely, sedentary behavior values measured with an accelerometer were almost 3 times higher compared to IPAQ.”
- A graphical abstract could improve the attractiveness of the paper
The graphical abstract is now included.
- The authors should stress among limitation that the study has been carried out in a specific country and setting, so that the results obtained can not immediately inferred to different settings and countries.
The limitations section of the paper now points out the fact that the results cannot be directly interpolated to other regions and/or demographics.
Once again, thank you for your valuable insight. We believe that the changes made will make the paper suitable for publication in Diagnostics.
Reviewer 2 Report
Comments and Suggestions for Authors
Thank you for the opportunity to review this manuscript, my considerations are below:
Title:
- Although the title implies that they are women, I suggest changing the term “breast cancer survivors” to “women breast cancer survivors”.
Abstract:
- I believe that adding subtopics (Background, objectives, methods, results and conclusion) will help improve the writing of the text and its understanding by readers. I suggest including these subtopics in the abstract.
- In the results, the authors report that there were measurements with significant differences, however, they do not include p-values, I suggest including all p-values of the reported outcomes.
- The conclusion of the study does not seem to answer the objective of the study, and brings conclusions about other outcomes. If the study presented several objectives, I suggest the authors mention in the objectives as primary, secondary objective... and in the conclusion section of the abstract, define the conclusion for each of these objectives, to make it clearer for readers.
- Diagnostics allows three to ten keywords, I suggest that authors expand their Keywords, as this can increase the chances of their article being found in a search for articles in the area and increase the chances of this article being cited.
Introduction:
- Well written, it presents a logical and temporal sequence of the topic, explaining in detail the problem of the study.
- However, the fourth paragraph is very long and disproportionate in relation to the others, I suggest dividing it into two.
- In the title, the authors use the term "breast cancer survivors", however, at the end of the introduction, they use the term "women who underwent treatment for breast cancer". Therefore, I suggest that the authors standardize the term throughout the study in relation to the volunteers in this study.
Methods:
- What is the design/type of the study?
- How was the sample recruited? Was there a calculation to estimate the sample size, or is it a convenience sample? The authors need to further detail how the sample for this study was recruited.
- What are the inclusion and exclusion criteria for the study? Despite some descriptions about the sample, these criteria are not clear in the study.
- For example: could the cancer be unilateral or bilateral? Could the treatment be either? Radiotherapy only? Or chemotherapy and radiotherapy? This may be important for the data, since chemotherapy is more disabling than radiotherapy. This needs to be clearer in the text.
- The text of the assessments is well written, detailing all stages of the study, explaining in detail how each of the stages of the assessments occurred.
- I suggest including photos of the accelerometer and the data provided by it, for a greater understanding of readers, who may not be familiar with this device.
Results:
- They are in accordance with the tables.
Discussion:
- When the authors mention the following text in the first paragraph of the discussion "The reluctance could be caused by concerns about possibility of upper extremity overload after lymphadenectomy as well as fatigue resulting from cancer treatment". This was mentioned by the participants, or this is what the authors believe, this needs to be clear in the text.
- In the eighth period of the discussion, the authors mention the following sentence: “Numerous research describes interventions aimed at increasing MVPA rates in cancer survivors [38, 39].” ...however, they only mention two references. Either the authors change the term “Numerous”, or they add more references to that phrase.
- I believe that the authors could mention the importance of the doctor after surgery, in clarifying doubts and encouraging survivors to perform physical activity, not only for patients, but for their children, as often, the decision to do whether or not physical activity will come from the child/caregiver of the elderly person. The role of the Physiotherapist must also be highlighted, as he is a professional who understands the disease and its consequences, understanding what lymphedema is and how to deal with it during exercise. I believe that doctors and physiotherapists can change this reality and I missed the text highlighting the importance of these professionals in changing this reality. I suggest adding.
- A final part of the discussion text is duplicated, please organize this better.
- I missed the discussion of implications for clinical practice and suggestions for future studies. These aspects contribute to more research being carried out and increase the chances of this article being cited. I suggest adding.
Conclusion:
- The conclusion must respond simply and directly to the objective of the study. Any other comments should be made at the end of the discussion. I suggest that the authors rewrite the conclusion to respond to the objective(s) of the study.
Author Response
Thank you for the opportunity to review this manuscript, my considerations are below:
Thank you for a thorough review of our manuscript. Below we provide comments to your recommendations.
Title:
- Although the title implies that they are women, I suggest changing the term “breast cancer survivors” to “women breast cancer survivors”.
The title has been changed as suggested.
Abstract:
- I believe that adding subtopics (Background, objectives, methods, results and conclusion) will help improve the writing of the text and its understanding by readers. I suggest including these subtopics in the abstract.
Although we believe that this form of abstract structuring would benefit its clarity, the Diagnostics guidelines for authors recommend not to include headings (“The abstract should be a single paragraph and should follow the style of structured abstracts, but without headings”).
- In the results, the authors report that there were measurements with significant differences, however, they do not include p-values, I suggest including all p-values of the reported outcomes.
P-values are now included in the results as suggested.
- The conclusion of the study does not seem to answer the objective of the study, and brings conclusions about other outcomes. If the study presented several objectives, I suggest the authors mention in the objectives as primary, secondary objective... and in the conclusion section of the abstract, define the conclusion for each of these objectives, to make it clearer for readers.
Your observation is accurate, the conclusions do not clearly answer the objective of the study. The objective was stated too generally. The conclusions are now corrected, however the abstract still includes only the main objective due to length limitation. The study objective section now includes the sentence “It was checked whether body weight had an impact on the level of physical activity and sedentary behavior”.
The secondary objectives were included:
- comparing the average moderate PA and MVPA values measured utilizing IPAQ questionnaire and accelerometer-based assessment.
- comparing the average sedentary behavior values measured utilizing IPAQ questionnaire and accelerometer-based assessment
- assessment of the relationship between physical activity, sedentary behavior, and general adiposity and adipose tissue distribution.
- Diagnostics allows three to ten keywords, I suggest that authors expand their Keywords, as this can increase the chances of their article being found in a search for articles in the area and increase the chances of this article being cited.
More key words were included: ActiGraph GT3X-BT, radiotherapy, chemotherapy.
Introduction:
- Well written, it presents a logical and temporal sequence of the topic, explaining in detail the problem of the study.
- However, the fourth paragraph is very long and disproportionate in relation to the others, I suggest dividing it into two.
The long paragraph is now divided.
- In the title, the authors use the term "breast cancer survivors", however, at the end of the introduction, they use the term "women who underwent treatment for breast cancer". Therefore, I suggest that the authors standardize the term throughout the study in relation to the volunteers in this study.
The nomenclature is now unified (“breast cancer survivors”).
Methods:
- What is the design/type of the study?
In the materials and methods section, we added information on the type of the study (original epidemiological clinical study).
- How was the sample recruited? Was there a calculation to estimate the sample size, or is it a convenience sample? The authors need to further detail how the sample for this study was recruited.
We included the information on the timing of the research (2022). We did not perform any calculations regarding the required sample size. Every patient visiting the rehabilitation outpatient clinic had been invited to the study. Patients who provided a conscious consent for participating in the study were included in the study.
- What are the inclusion and exclusion criteria for the study? Despite some descriptions about the sample, these criteria are not clear in the study.
- For example: could the cancer be unilateral or bilateral? Could the treatment be either? Radiotherapy only? Or chemotherapy and radiotherapy? This may be important for the data, since chemotherapy is more disabling than radiotherapy. This needs to be clearer in the text.
Information about inclusion and exclusion criteria is now included. We would like to point out that we are aware of the necessity of narrowing down the recruitment criteria, and currently we are continuing our research taking into account the age groups and time elapsed since the surgery. In 2022 the fear of COVID-19 discouraged women from taking part in the study. In the future studies, the patients will be divided based on the type of surgical treatment. We agree that every method of cancer treatment leads to different adverse events, however, their intensity depends on the dose, duration of treatment and others. Multimorbidity, often found in our patients, is also an important factor influencing their physical activity. Patients who underwent chemotherapy, and/or radiation therapy were included in our study. We are aware of the fact that expanding the studied population will be needed to standardize it, thus further research is underway.
- The text of the assessments is well written, detailing all stages of the study, explaining in detail how each of the stages of the assessments occurred.
- I suggest including photos of the accelerometer and the data provided by it, for a greater understanding of readers, who may not be familiar with this device.
A graphical abstract is now added, including the photos of the accelerometer.
Results:
- They are in accordance with the tables.
Discussion:
- When the authors mention the following text in the first paragraph of the discussion "The reluctance could be caused by concerns about possibility of upper extremity overload after lymphadenectomy as well as fatigue resulting from cancer treatment". This was mentioned by the participants, or this is what the authors believe, this needs to be clear in the text.
The mentioned sentence is the authors’ conclusion. It comes from many years’ experience in working with cancer patients and observation of the problem of lymphatic edema. Literature supports this notion
(Murray, J; Perry, R; Pontifex, E; Selva-Nayagam, S; Bezak, E; Bennett, H. The impact of breast cancer on fears of exercise and exercise identity. Patient Education and Counseling, 2022; 105, 7, 2443-2449.
Jammallo, L; Miller, C; BS, Horick, N. at. all. Factors Associated With Fear of Lymphedema After Treatment for Breast Cancer. Oncology Nursing Forum, 2014, 41(5), 473–483. doi: 10.1188/14.ONF.473-483.
Lin, H; Kuo, Y; Tai, W; Liu, H. Exercise effects on fatigue in breast cancer survivors after treatments: A systematic review and meta-analysis. International Journal of Nursing Practise, 2022, 28,4. https://doi.org/10.1111/ijn.12989).
Later part of the discussion also highlights that physical activity decreases fatigue (“It has been established that undertaking PA after diagnosis of breast cancer decreases fatigue and other adverse effects of cancer treatment, along with improving life quality and overall fitness of cancer survivors”).
- In the eighth period of the discussion, the authors mention the following sentence: “Numerous research describes interventions aimed at increasing MVPA rates in cancer survivors [38, 39].” ...however, they only mention two references. Either the authors change the term “Numerous”, or they add more references to that phrase.
More literature supporting this sentence was added.
- I believe that the authors could mention the importance of the doctor after surgery, in clarifying doubts and encouraging survivors to perform physical activity, not only for patients, but for their children, as often, the decision to do whether or not physical activity will come from the child/caregiver of the elderly person. The role of the Physiotherapist must also be highlighted, as he is a professional who understands the disease and its consequences, understanding what lymphedema is and how to deal with it during exercise. I believe that doctors and physiotherapists can change this reality and I missed the text highlighting the importance of these professionals in changing this reality. I suggest adding.
- I missed the discussion of implications for clinical practice and suggestions for future studies. These aspects contribute to more research being carried out and increase the chances of this article being cited. I suggest adding.
According to your recommendations the role of the physician and physiotherapist in promoting physical activity is now pointed out. Implications for clinical practice and directions for future research are now added.
“A crucial role in the education of patients and their families is attributed to the physican and the physiotherapist. At Holycross Cancer Center, where the research was conducted, instructions on anti-swelling prophylaxis and physical exercise tutorial are given one day prior to surgery. This education, along with exercise and self-massage tutorials are continued throughout the patient’s hospitalization. Additionally, patients receive an informative booklet containing essential guidance. Implementing a program to monitor physical activity in specific periods of time could be beneficial. In case of reduced PA, the physician and physiotherapist could interview the patient in search of the cause of this situation and intervene if necessary. It is our conclusion that PA level depends not only on the education carried out by the personnel, but also on the pa-tient’s exercise habits before cancer. Patients’ well-being during various phases of treatment also holds significant importance. Our future research will focus on PA of breast cancer survivors examining how it varies over time following surgery and in response to different treatment modalities.”
- A final part of the discussion text is duplicated, please organize this better.
The final, duplicated part of the text is now corrected.
Conclusion:
- The conclusion must respond simply and directly to the objective of the study. Any other comments should be made at the end of the discussion. I suggest that the authors rewrite the conclusion to respond to the objective(s) of the study.
The conclusions were edited according to your recommendations. We pointed them out, while referring to study objectives.
Once again, we would like to thank you for your valuable insight. We believe that it will allow us to prepare better and more complex studies in the future.
Round 2
Reviewer 2 Report
Comments and Suggestions for Authors
Congratulations to the authors who did a good job. All my requests were met, and I believe that after the changes the article is clearer and more understandable for readers.